# Generation of a Triple-Shuttling Vector and the Application in Plant Plus-Strand RNA Virus Infectious cDNA Clone Construction

**DOI:** 10.3390/ijms24065477

**Published:** 2023-03-13

**Authors:** Chenwei Feng, Xiao Guo, Tianxiao Gu, Yanhong Hua, Xinjian Zhuang, Kun Zhang

**Affiliations:** 1Department of Plant Pathology, College of Plant Protection, Yangzhou University, Yangzhou 225009, China; 2Joint International Research Laboratory of Agriculture, Agri-Product Safety of Ministry of Education of China, Yangzhou University, Yangzhou 225009, China; 3Plant Protection Research Institute, Guangdong Academy of Agricultural Sciences/Guangdong Provincial Key Laboratory of High, Technology for Plant Protection, Guangzhou 510640, China; 4Jiangsu Key Laboratory for Microbes and Functional Genomics, Jiangsu Engineering and Technology Research Center for Microbiology, College of Life Sciences, Nanjing Normal University, Nanjing 210023, China

**Keywords:** triple-shuttling vector, plant virus infectious clone, homologous recombinant (HR)-based cloning, yeast

## Abstract

Infectious cloning of plant viruses is a powerful tool for studying the reverse genetic manipulation of viral genes in virus–host plant interactions, contributing to a deeper understanding of the life history and pathogenesis of viruses. Yet, most of the infectious clones of RNA virus constructed in *E. coli* are unstable and toxic. Therefore, we modified the binary vector pCass4-Rz and constructed the ternary shuttle vector pCA4Y. The pCA4Y vector has a higher copy number in the *E. coli* than the conventional pCB301 vector, can obtain a high concentration of plasmid, and is economical and practical, so it is suitable for the construction of plant virus infectious clones in basic laboratories. The constructed vector can be directly extracted from yeast and transformed into *Agrobacterium tumefaciens* to avoid toxicity in *E. coli*. Taking advantage of the pCA4Y vector, we established a detailed large and multiple DNA HR-based cloning method in yeast using endogenous recombinase. We successfully constructed the Agrobacterium-based infectious cDNA clone of ReMV. This study provides a new choice for the construction of infectious viral clones.

## 1. Introduction

Plant virus infectious clones are a powerful tool for reverse genetic manipulation of viral genes in studying virus–host plant interactions and have contributed to a deeper understanding of virus life cycle and pathogenesis [1]. Infectious clones of various plus-strand RNA viruses have been successfully constructed using controlled transcription of viral full-length genomic RNAs by T7 and 35S promoters in vitro and in vivo since the infectious clone of the brome mosaic virus was first generated in 1984 [1,2,3,4]. Owing to the tedious sub-cloning and sequential cloning steps, the traditional digestion–ligation methods, which take advantage of *E. coli* to amplify the constructed infectious clones, are laborious, time-consuming, and require many molecular reagents. Obtaining infectious clones of many plant viruses using traditional methods is difficult because of the large viral genomic RNA/DNA size, unexpected deletions, and toxicity to *E. coli* growth [5,6,7]. In contrast, homologous recombination (HR)-based cloning depends on yeast (*Saccharomyces cerevisiae*) and is a highly efficient and cost-effective method for the seamless assembly of DNA fragments, especially in potyvirus infectious clone construction [7,8]. Taking advantage of HR in yeast cells, which is referred to as gap-repair cloning, could achieve multiple DNA fragment assembly with 20–30 bp homologous ends [4]. This method does not require the use of enzymes or common restriction sites. The co-transformation of yeast cells with linearized vectors and the insertion of DNA fragments (with homologous sequences in the 5′ and 3′ regions of these two components) efficiently form a circular plasmid [4,9]. Owing to its high transformation efficiency, simplicity, and capacity for complex DNA assembly, HR in yeast has been used in high-throughput plasmid construction, genome synthesis, and genome engineering [4]. HR-based cloning in yeast has already been successfully applied to the construction of several plant RNA virus infectious clones for agrobacteria-based infiltration [8,9,10]. 

*Cucumber mosaic virus* (CMV) is a member of the genus *Cucumovirus* and has a wide host range worldwide. CMV can be transmitted by different species of aphids (>75 species) in a non-persistent manner, causing large losses to vegetables, legumes, and ornamentals [11]. Its genome contains three positive-sense single-stranded RNAs, designated RNA1 (~3350 nucleotides), RNA2 (~3050 nucleotides), and RNA3 (~2200 nucleotides), each separately encapsidated in icosahedral virions [12]. Thus, each mature CMV consists of three types of spherical particles. RNA1 encodes the 1a protein, necessary for viral genomic RNA replication. RNA2 encodes two proteins, 2a and 2b. The 2a protein is involved in viral genome replication and association with the 1a protein, whereas 2b counters the host’s post-transcriptional gene silencing, thereby allowing the CMV to continue to invade and infect young developing tissues. The 3a protein encoded by RNA3 is a movement protein that is essential for virus cell-to-cell movement, and mutations in this protein have been identified as related to host-specific differences in movement efficiency. The 3b protein is the coat protein encoded by the sub-genomic RNA, referred to as RNA4. The coat protein is the only protein associated with virus particles and is the sole determinant of transmission by aphid vectors [13].

*Tobacco mosaic virus* (TMV), a member of a large group of viruses within the genus *Tobamovirus*, has single-stranded genomic RNA (~6400 nucleotides) and rod-shaped viral particles (virions) [14]. *Youcai mosaic virus* (YoMV) and *Rehmannia mosaic virus* (ReMV), which were previously isolated from *Raphanus sativus* and *Rehmannia glutinosa*, respectively [15,16], also belong to the genus *Tobamovirus*. *Tobamoviruses* are the most important viruses in crops that are not transmitted by invertebrate vectors. Virus-carrying seeds and debris from previous crops in the soil usually serve as the primary infection sources, with subsequent spread within the crops through direct contact between plants and mechanical transfer of virus on the hands and equipment of those working with the crops [17]. Rape virus disease occurred in rape-producing areas all over the country, among which the winter rape areas in North China, Southwest China, and Central China were the most serious [16]. In the long-term asexual reproduction process, ReMV and other viruses will accumulate and aggravate the damage, causing a serious impact on the yield and quality of groundnut [15].

In the present study, we constructed a triple-shuttling vector between *E. coli*, *Agrobacterium*, and yeast. Taking advantage of the constructed vector, we developed a simplified method for the construction of a positive-strand plant RNA virus infectious cDNA clone by utilizing the yeast HR capacity for large-insert cloning and the advantage of agro-infiltration. We successfully constructed functional infectious full-length CMV-Fny, YoMV, and ReMV cDNA clones using several fragments assembled by yeast HR in a relatively rapid and single step. Moreover, the assembled vectors can be isolated from the yeast and directly transformed into *A. tumefaciens* for subsequent agro-infiltration. This procedure avoids the vector propagation step in *E. coli* and circumvents concurrent plasmid instability issues. The obtained infectious full-length cDNA clones of CMV-Fny, YoMV, and ReMV can also be stably maintained in *E. coli* and subsequently genetically engineered. The pCA4Y vector has a higher copy number in the *E. coli* than the conventional pCB301 vector, hence this is more conducive to virus replication and increased plasmid concentration. At the same time, this method is more economical and does not require the purchase of expensive biochemical reagents, making it more suitable for the construction of plant virus infectious clones in basic laboratories.

## 2. Results

### 2.1. Construction of the S. cerevisiae–E. coli–Agrobacterium Shuttle Vectors and Development of HR-Based Cloning in Yeast

*Agrobacterium* binary vectors have the plasmid replication origin element and the corresponding biotic selection marker, which is capable of self-replication in both *Agrobacterium* and *E. coli* but lacks the DNA elements for multiplication in yeast cells. To obtain a vector that can shuttle into different hosts (*S. cerevisiae–E. coli–Agrobacterium*), we chose the binary vector pCass4-Rz as the backbone, which contains the strongly driven 35S promoter, HDV antigenomic ribozyme, and NOS terminator elements (Figure 1). Sequences containing the yeast 2 μ replication origin and the gene encoding tryptophan (*TRP1*) autotrophic marker were cloned by PCR from the commercial plasmid pGBKT7 (Accession No. LT726927.1) (S Appendix A) and inserted into the *Sac* II endonuclease sites of the binary pCass4-Rz vector, which generated the *S. cerevisiae*–*E. coli*–*Agrobacterium* shuttle vector, pCA4Y (Figure 1). The constructed vector is capable of replication in yeast and allows the yeast to grow well in the tryptophan drop-out medium.

Taking advantage of the pCA4Y vector, we established a detailed large and multiple DNA HR-based cloning method in yeast using endogenous recombinase (Figure 2). The detailed operation procedures are illustrated. First, we linearized the vector using the existing *Stu* I/*Bam* HI endonuclease. Specific primer pairs were designed with a forward adaptor and reverse adaptor (Figure 2), and the target DNA fragment was then amplified using the designed primers. Moreover, the target DNA fragment and linearized vector were mixed at a molar ratio (2:1) and co-transformed into the yeast strain *YPH500*. The positive transformants grew well in a tryptophan drop-out medium with kanamycin resistance.

### 2.2. Construction of the CMV-Fny cDNA Infectious Clones Using the Triple-Shuttle Vector through Yeast Homologous Recombination

We cloned full-length CMV-Fny cDNA to verify the HR-based cloning efficiency for multiple fragment assembly in plant-positive RNA virus infectious cDNA synthesis. The CMV-Fny genome contained three RNA fragments designated RNA1, RNA2, and RNA3 (Figure 3A). According to the developed HR-based cloning method, we amplified the cDNAs corresponding to RNA1, RNA2, and RNA3 using pairs of primers (Figure 3B). Each amplified cDNA was mixed with linearized pCA4Y and transformed into the yeast strain *YPH500*. The positive clone was selected by plates with tryptophan drop-out medium with kanamycin resistance and then isolated from the yeast. We named the positive cDNA clones corresponding to the CMV-Fny of each genomic RNA pCA4Y-RNA1, pCA4Y-RNA2, and pCA4Y-RNA3. The target plasmids were multiplied by yeast strain *YPH500* and then subjected to validation by PCR. The PCR results showed that each plasmid contained the target cDNA that corresponded to their corresponding genomic RNA, and all plasmids contained the yeast 2 μ fragment (Figure 3C). The target plasmids were further transformed into the *Agrobacterium tumefaciens* strain GV3101. *Agrobacterium-*mediated inoculation of CMV-Fny to *N. benthamiana* plants was performed using two types of CMV-Fny infectious clones (pCB301-CMV-Fny and pCA4Y-CMV-Fny), the pCB301 vector was granted by Dr. Xiaorong Tao (Nanjing Agricultural University) [18]. After 14 days post-inoculation (dpi), the leaves showed yellowing, rolling, and mosaic symptoms, indicating that the plants were infected by CMV-Fny (Figure 3D). The leaves were collected for total protein extraction. Protein immunoblot analyses were performed using the CMV-Fny coat protein-specific antibodies, and the results showed that the pCA4Y-CMV-Fny positive *Agrobacterium-*inoculated plants were infected by CMV-Fny, as well as the positive control (pCB301-CMV-Fny-positive *Agrobacterium*-infiltrated plant) (Figure 3E, bottom). The RT-PCR results also showed that the pCA4Y-CMV-Fny-positive *Agrobacterium*-inoculated plants appeared in similar size bands to pCB301-CMV-Fny-positive *Agrobacterium*-infiltrated plants on the gel, indicating that the pCA4Y-CMV-Fny-positive *Agrobacterium*-inoculated plants were infected with CMV-Fny (Figure 3E, upper). Taken together, these results strongly demonstrate that the constructed CMV-Fny cDNA clone pCA4Y-CMV-Fny has infectious activity against *N. benthamiana* and that the yeast HR-based cloning method was effective.

### 2.3. Construction of the YoMV Full-Length cDNA Infectious Clones 

To further validate whether the developed yeast HR-based cloning method could be used for cDNA infectious clone construction of other positive plant viruses, we selected YoMV isolated from the *R. sativus* planted in Henan Province. The YoMV genome is a single positive-strand RNA of approximately 6.3 kilo-nucleotides in length (Figure 4A), which is hard to clone using the traditional molecular cloning method by the endonuclease. Hence, we divided the genome sequence into two fragments and amplified respective cDNAs by RT-PCR from the total cDNAs of *R. glutinosa*. These two fragments were 2991 and 3335 bp in length, respectively (Figure 4B). Similarly, we mixed fragments 1 and 2 with the linearized pCA4Y vector and co-transformed it with yeast *YHP500*. PCR amplification of the yeast 2 μ fragment, viral fragment 1, and fragment 2 was performed to validate the effectiveness of HR-based cloning using the developed method in yeast (Figure 4C). The results showed that the pCA4Y-YoMV plasmid was successfully constructed (Figure 4C). Furthermore, we transformed the positive pCA4Y-YoMV plasmid into *Agrobacterium tumefaciens* strain GV3101 and inoculated the obtained positive clone into *N. benthamiana*. After 5 dpi, yellowing, rolling, mosaic, and slight necrosis appeared on the upper leaves of the inoculated plants, similar to the pCB301-YoMV-positive *Agrobacterium*-inoculated plants (positive control) (Figure 4D), the pCB301-YoMV vector was granted by Dr. Zhenggang Li (Guangdong Academy of Agricultural Sciences, Guangzhou, China) [16]. These results indicated that yeast HR-based YoMV infectious cDNA construction was successful. Moreover, we collected the upper symptomatic leaves and performed Western blotting and RT-PCR using YoMV coat protein-specific antibodies and CP gene-specific primers, respectively. The results showed that all pCA4Y-YoMV-positive *Agrobacterium*-inoculated plants were infected by YoMV, as well as the positive control (Figure 4E). In summary, these results demonstrated that the yeast HR-based YoMV infectious cDNA construction was effective and successful, and the constructed pCA4Y could be used for the efficient construction of positive single-strand plant virus cDNA clones.

### 2.4. Construction of the ReMV Full-Length cDNA Infectious Clones

To further test the efficiency of the yeast HR-based construction of plant virus cDNA clones, ReMV was selected as the target. ReMV belongs to the same genus as YoMV and has a similar genome organization (Figure 5A). Our previous study showed that *R. glutinosa* collected from Henan Province in China was infected with both *Cucurbit chlorotic yellow virus* (CCYV) and ReMV [19]. Hence, the total cDNA of *R. glutinosa* used above was treated as a template, and RT-PCR was performed. Similarly, we divided the ReMV genome into three segments and designed three primer pairs. Using these primers and the total cDNA templates, we amplified cDNA fragments 1 (2159 bp), 2 (2161 bp), and 3 (2121 bp) (Figure 5B). Next, we performed yeast HR-based construction of a full-length cDNA clone of ReMV. The pCA4Y-ReMV plasmid was further validated by PCR using primers corresponding to the viral cDNA fragments 1, 2, and 3 and yeast 2 μ element. The results showed that all fragments could be amplified from the pCA4Y-ReMV plasmid, indicating that the ReMV full-length cDNA clone was successfully constructed (Figure 5B). To test whether the ReMV full-length cDNA clone was infectious in *N. benthamiana*, we transformed the plasmid into *Agrobacterium tumefaciens* strain GV3101 and inoculated the *Agrobacterium* into *N. benthamiana* by infiltration. After 5 dpi, the leaves appeared as yellowing, curling, and mosaic, even with local cell necrosis (Figure 5D), which implied that the inoculated plants were infected by ReMV. To determine whether these symptoms were caused by ReMV, upper symptomatic leaves were collected for total protein and RNA extraction. RT-PCR and Western blotting were performed using pair of primers and antibodies specific to coat protein-coding sequence and CP, respectively. (Figure 5E). The results indicated that the inoculated plants were infected with the constructed ReMV cDNA clones as well as the positive control (pCB301-ReMV), the pCB301-ReMV vector was constructed this time. Taken together, the results demonstrated that the constructed triple-shuttle plasmid pCA4Y and the developed yeast HR-based multiple fragment cloning were effective and suitable for the construction of positive single-stranded plant virus infectious cDNA.

## 3. Discussion

The key to the generation of biologically active plant-positive single-stranded RNA virus infectious clones is to place the cDNAs between a suitable promoter and a precise cutting activity ribozyme for the authentication of both the 5′ and 3′ termini of the transcribed viral genomic RNAs [1,20]. To successfully construct an RNA virus infectious clone, sophisticated designs, a series of subcloning steps using traditional restriction endonuclease digestion and fragment ligation, and a variety of expensive molecular cloning enzymes and reagents are required. New strategies, such as in-fusion cloning and Gibson Assembly, have also been used in the construction of virus infectious clones [7,21,22,23]. We supplied HR-based cloning vector construction methods, in which the fragments flanking homologous ends are seamlessly assembled in vitro, allowing for simple, straightforward, and efficient vector construction with low cost. This gives some basic laboratories a cheaper alternative. Multiple DNA fragments, with 20–30 bp homologous end flanking, could prime the scarless assembly in yeast using the endogenous repair machinery. Various studies have demonstrated the efficiency and capacity of HR-based cloning in the construction of infectious cDNA clones of plant plus-strand RNA viruses [7,8,9,24,25,26,27].

In the present study, we constructed a triple-shuttle vector pCA4Y that could replicate in *S. cerevisiae*, *E. coli*, and *A. tumefaciens* (Figure 1). We also assessed the yeast HR capacity in multiple fragment assembly in one step (Figure 2) and the direct transferring ability of virus cDNA into host plant genomic DNA by *Agrobacterium* infection. Based on this, we successfully constructed three positive single-strand plant RNA virus infectious cDNA clones in a short time, namely, CMV-Fny, YoMV, and ReMV (Figure 3, Figure 4 and Figure 5). The observed symptoms, molecular detection, and serological detection simultaneously demonstrated that the plant viruses’ infectious cDNA clones based on yeast HR cloning have biological activity and are successfully rescued (Figure 3D,E, Figure 4D,E, and Figure 5D,E). The yeast HR-based cloning method is simple, efficient, and requires no extraordinarily well-trained operators, expensive reagents, or professional equipment. Additionally, it overcomes the toxicity of coding products caused by similar polycistron strategies in both plant viruses and bacteria during target plasmid multiplication in *E. coli*. The plasmid with biological activity could be recovered directly from the yeast and then transformed into *Agrobacterium*, skipping the propagation step in *E. coli*, which avoids the killing role of the viral gene sequences and encoding products in the cells. The majority of infectious cDNA clones of RNA viruses constructed in *E. coli* are unstable and toxic, such as in animal viruses, the NS1 (non-structural protein 1) of Zika virus, the genus of the *Flavivirus*, is toxic and can be mutated easily if constructed within *E. coli* [28,29]. In addition, in plant viruses, *Potato virus Y* and other members of the family *Potyviridae* are well known for their difficulty in infectious cDNA clones constructing, mainly as a result of toxicity issues [6,23,30]. Dengue virus type 4 [31], RNA2 of *Beet necrotic yellow vein virus* (BNYVV) [32], *Pea early browning virus* (PEBV) [33], *Eggplant mosaic virus* (EMV) [34], and other viruses have similar problems.

Adding intron sequences to the cloned cDNA with respect to viral genomic RNAs is a common strategy, which could stabilize the cloned viral cDNA in *E. coli* [35,36,37,38]. However, the number of introns that should be used and the sites for intron insertion into the respective viral cDNA clones often have no strict rules and standards, which leads to the successful construction of infectious cDNA clones of a few viruses [4]. Infectious cDNA clones of numerous plant viruses, especially those that cause economically important crop diseases, are unavailable [39,40]. Using the yeast HR-based cloning method, Sun et al. (2017) constructed an intronless PVY cDNA clone that could be obtained by rapid assembly and is highly infectious [4]. Similarly, highly infectious cDNA clones of the *Sonchus yellow net virus*, *Soybean mosaic virus*, and *Chili veinal mottle virus* were successfully constructed [4,8,9]. Hence, the yeast HR-based cloning method can be used to construct intronless plant virus cDNA clones.

*Rehmannia glutinosa* is an important traditional Chinese medicinal plant, and its roots can be used to treat hemoptysis, anemia, and gynecological disease [41]. Planting *R. glutinosa* is key for the government and farmers to eliminate poverty in middle China under the current background of rural revitalization [15]. *Raphanus sativus* L., commonly named radish, is an important species cultivated in most of the temperate regions of the world and severed as an ancient kitchen garden vegetable. However, the viral disease often causes remarkable loss in the yield and quality of *R. sativus* and *R. glutinosa*, and at least seven viruses have been reported in *R. glutinosa* in China [42]. In this study, we successfully constructed three *Agrobacterium* delivery system-based intron-free plant virus cDNA clones (Figure 3, Figure 4 and Figure 5). Among them, the YoMV and ReMV cDNA clones, which were isolated from *R. glutinosa* in Henan Province, were successfully rescued and were highly infectious in *N. benthamiana* (Figure 3D, Figure 4D, and Figure 5D). Further investigations are required to determine the infection of ReMV in *R. glutinosa.* Our study provides the reverse genetic tools for YoMV and ReMV for a deeper investigation of the pathogenesis-related molecular mechanisms involved in the interactions between plant viruses and host plants, which will be beneficial for the breeding of *R. sativus* and *R. glutinosa* with high resistance and yield in the future.

## 4. Method and Materials

### 4.1. Vector Information and Plant Culture Condition

In the present study, three plant-positive single-stranded viruses were selected: CMV-Fny, YoMV, and ReMV. We obtained cDNA clones of CMV-Fny but not the other two viruses. YoMV and ReMV have been isolated from *R. sativus* and *R. glutinosa*, a traditional Chinese medicinal plant. We chose the *E. coli*–*Agrobacterium* dual shuttle vector as the original vector. All revisions or insertions were based on vectors.

*N. benthamiana* was cultivated in the greenhouse of the Plant Molecular Laboratory at Yangzhou University under conditions of 24 ± 1 °C with a photoperiod of 16/8 h, which included 16 h of four-level light and 8 h of dark treatment.

### 4.2. Candidate Virus and Viral RNA

The plasmids of CMV-Fny cDNA infectious clones were obtained from Professor Xianbing Wang from China Agriculture University (which comes from Dr. Ding’s Lab). YoMV- and ReMV-infected *R. sativus* and *R. glutinosa* leaves were also harvested from a Chinese herbal medicine garden located in Henan Province. For CMV-Fny, cDNA fragments with specific adaptors were obtained by PCR, with the template being the preserved plasmids of CMV-Fny cDNA infectious clones, and the pairs of primers are listed in Appendix A. For YoMV and ReMV, total RNAs were extracted from the viruses-infected *R. sativus* and *R. glutinosa* leaves, respectively, and reverse transcription was performed using the reverse primers listed in Appendix A. The different cDNAs obtained with respect to these two viruses (YoMV and ReMV) were used as templates, and the corresponding pairs of primers were selected for the amplification of cDNAs of different lengths.

### 4.3. Construction of the Triple-Shuttling Vector in S. cerevisiae, E. coli, and Agrobacterium

To obtain the yeast 2 μ replication origin element and the autotrophic marker sequences, we used the commercial pGBKT7 plasmid as a template and the pairs of primers pGBK-2686F1/pGBK-2686R1 to amplify the target fragment containing the specific adapters. Restriction endonuclease *Sac* II was used to linearize the dual vector pCass4-Rz. Then, the obtained yeast 2 μ replication origin element, autotrophic marker sequences, and purified linearized pCass4-Rz were mixed and assembled in vitro using Gibson Assembly^®^ Master Mix (New England Biolabs, Beijing, China, Cat No. E2611S). The products of the reaction were transformed into yeast strain *YPH500*. Tryptophan drop-out medium containing kanamycin was used to screen for positive clones. The target plasmid pCA4Y was isolated and purified by the Yeast Plasmid Extraction Kit (Solarbio, Beijing, China, Cat no. D1160). After the experimental test, the triple-shuttle vector pCA4Y was most suitable propagated in *E. coli* strain JM110.

### 4.4. Assembly of CMV-Fny, YoMV, and ReMV cDNA Clones by Yeast Homologous Recombination

For CMV-Fny, we amplified three cDNA fragments using the specific primer pairs listed in Appendix A. The three cDNAs corresponded to each strand of the CMV-Fny genomic RNAs. We mixed each cDNA fragment with *Stu* I/*Bam*H I linearized pCA4Y with a mole ratio of 2/1 in vitro and transformed it into the yeast strain *YPH500* using the Yeast Transformation Kit (Coolaber, Beijing, China, Cat no. SK2400-200T). Three positive CMV-Fny cDNA clones for the three genomic RNA were obtained using a tryptophan drop-out medium with kanamycin selection. 

For YoMV, we divided the full-length cDNA into two overlapping fragments: one 2991 bp in length and the other 3335 bp in length. The specific adapters were added by RT-PCR. Similarly, purified fragment 1 (2991 bp), fragment 2 (3335 bp), and linearized pCA4Y were mixed in a molar ratio of 2:2:1 in vitro and then transformed into yeast. One positive cDNA clone with respect to full-length ReMV genomic RNA without scars was obtained. 

For the ReMV, the full-length cDNA was divided into three fragments: the first fragment was 2159 bp, the second fragment was 2161 bp, and the third fragment was 2121 bp. Similar procedures were performed for YoMV full-length cDNA clone construction.

### 4.5. Plasmid Purification from Yeast and Propagation

We obtained the three plant viruses (CMV-Fny, YoMV, and ReMV). They contain five destination plasmids based on the constructed triple-shuttle vector pCA4Y, which includes pCA4Y-CMV-Fny-RNA1, pCA4Y-CMV-Fny-RNA2, pCA4Y-CMV-Fny-RNA3, pCA4Y-YoMV, and pCA4Y-ReMV. They were propagated in yeast, and then the five vectors were extracted and purified by the Yeast Plasmid Extraction Kit (Solarbio, Beijing, China, Cat no. D1160).

### 4.6. Agrobacterium-Mediated Infiltration of N. benthamiana Leaves

The constructed pCA4Y-CMV-Fny-RNA1, pCA4Y-CMV-Fny-RNA2, pCA4Y-CMV-Fny-RNA3, pCA4Y-YoMV, and pCA4Y-ReMV were transformed into *Agrobacterium tumefaciens* strain GV3101 using the freeze–thaw method [43]. Positive clones of *Agrobacterium tumefaciens* strain GV3101 were selected for kanamycin and tetracycline antibiotics and validated by PCR. *Agrobacterium tumefaciens* transformed with the pCA4Y empty plasmid was used as a negative control. For CMV-Fny, *A. tumefaciens* containing pCA4Y-CMV-Fny-RNA1, pCA4Y-CMV-Fny-RNA2, and pCA4Y-CMV-Fny-RNA3 were cultured in Luria Bertani liquid culture medium and suspended in suspension buffer (2 mL 1 M 2-Morpholinoethanesulphonic acid, 2 mL 1 M MgCl_2_, 150 µL 150 mM Acetosyringone), as described previously [44]. The suspensions were then mixed to obtain a total OD_600_ of 0.8. After a 2 h incubation at 28 °C, the suspension was infiltrated into *N. benthamiana*. Similarly, the OD_600_ of suspensions of *A. tumefaciens* containing pCA4Y-YoMV and pCA4Y-ReMV was adjusted to 0.5. 

### 4.7. RT-PCR and Western Blot Detection of the Target Virus

For RT-PCR detection, the upper symptomatic leaves of *N. benthamiana* were harvested for total RNA extraction using Trizol reagent (Invitrogen, ThermoFisher, Shanghai, China). Reverse transcription was performed using reverse pairs of primers for sequences encoding the coat protein (CP) of each virus (Appendix A). PCR was performed to amplify the CP fragments of three viruses (CMV-Fny, YoMV, and ReMV). For the Western blot assay, extraction of total protein from 0.2 g of symptomatic *N. benthamiana* leaves using liquid nitrogen grinding. Then, a total of 200 µL 2X SDS protein loading buffer (100 mM Tris-HCl, 20% glycerin, 4% sodium dodecyl sulfate, 0.2% bromophenol blue, 5% β-mercaptoethanol) was added as previously described [44]. The extract was vortexed and placed in boiling water for 10 min. Protein samples were loaded onto a 12.5% SDS-polyacrylamide gel (SDS-PAGE) for electrophoresis at 80 V for 2 h. Proteins separated on the gel were transferred onto a nitrocellulose membrane at 200 mA for 2 h. Specific antibodies against the CP proteins of CMV-Fny, YoMV, and ReMV were used to detect the infection of the target viruses. The secondary antibody used was alkaline phosphatase (AP)-conjugated antirabbit immunoglobulin (AP-A) (Sangon Biotech, CA, USA).

## Figures and Tables

**Figure 1 ijms-24-05477-f001:**
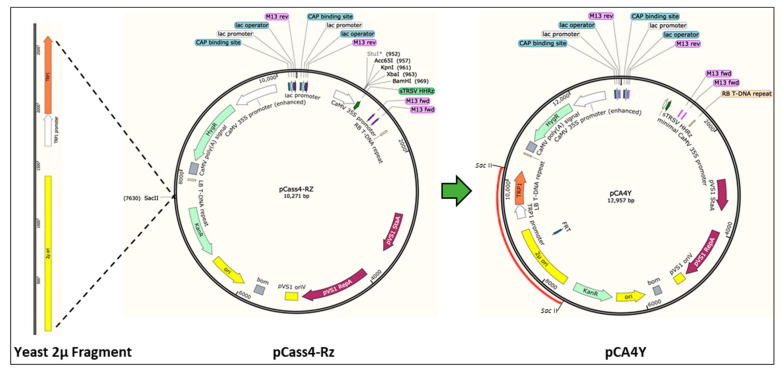
Schematic representation of the construction flows of the yeast–*E. coli*–*Agrobacterium* shuttle vector pCA4Y. Briefly, we amplified the yeast 2 μ replication origin and the tryptophan (TRP1) autotrophic marker gene fragment from the commercial pGBKT7 plasmid (Accession No. LT726927.1) and inserted the *Sac* II site of the binary vector pCass4-Rz.

**Figure 2 ijms-24-05477-f002:**
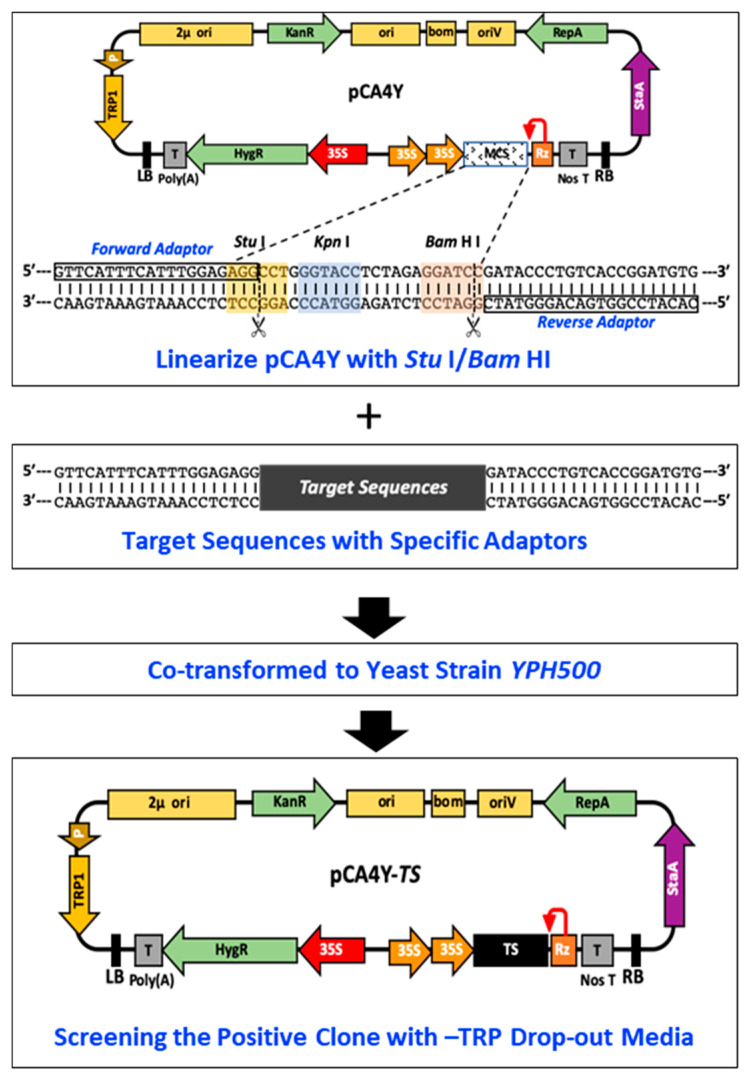
Schematic illustration of the construction procedures based on yeast homologues recombination. First, we should linearize the pCA4Y by *Stu* I and *Bam* HI double-digestion (vector). Then, to obtain the target fragment by amplification the fragments with specific adaptor shown above (fragments). If the construction involved multiple fragments ligation, each pair of adjacent fragments should have a homologues sequence with at least 20 bp length. Second, the prepared fragments and vector were co-transformed to the yeast strain *YPH500*, and the positive clones were screened by the TRP1 drop-out medium with kanamycin and PCR with specific primers.

**Figure 3 ijms-24-05477-f003:**
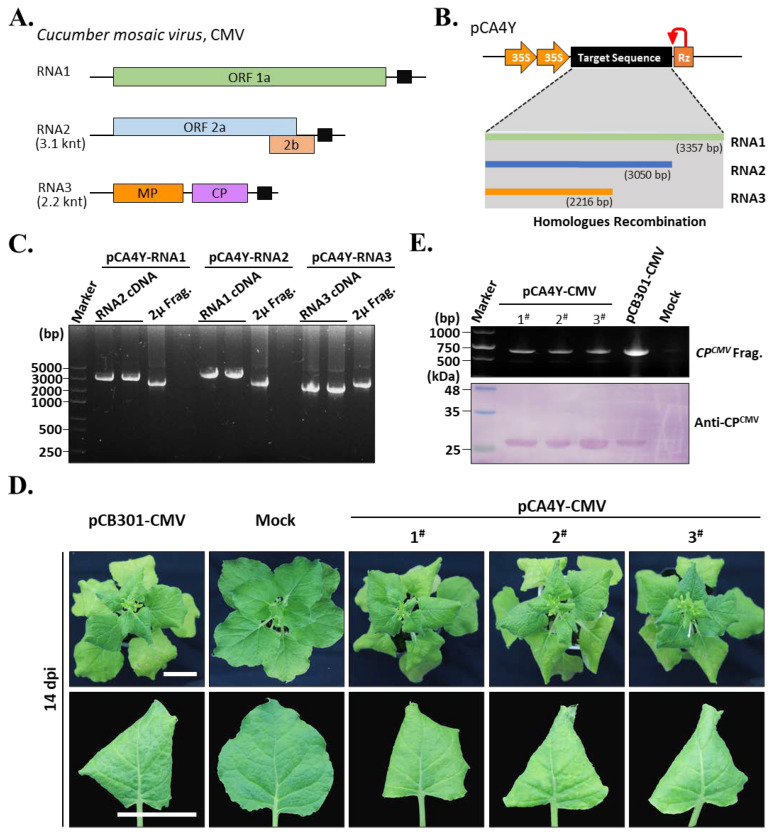
Construction of the *Cucumber mosaic virus* (CMV-Fny) infectious cDNA clone using the established triple-shutting vector-based yeast HR recombination. (**A**). Genome organization of the CMV-Fny. ORF1 and ORF2 are associated together and formed the viral replicase. 2b is a classical viral encoded gene silencing suppressor. RNA3 encodes the movement protein (MP) and coat protein (CP). (**B**). Illustration of the detailed construction strategies for three cDNAs that correspond to the CMV-Fny genomic RNAs. (**C**). PCR validation of the positive CMV-Fny infectious cDNA clones with specific pairs of primers listed in Appendix A. 2 μ Frag. was the short name of the yeast 2 μ replication origin and the TRP1 autotrophic marker gene fragment. (**D**). Symptoms of the CMV-Fny infectious cDNA clones-inoculated *Nicotiana benthamiana* plants after 14 dpi. The pCB301-CMV-Fny-inoculated plants were treated as the positive control, and the blank agrobacterium-infiltrated plants were served as mock. Numbers 1^#^, 2^#^, and 3^#^ represent the three independent inoculated plants. Bar indicates 3 cm. (**E**). RT-PCR and Western blot detect the infectivity of the newly constructed CMV-Fny infectious clones. Specific pairs of primers amplification of the *CP* coding region or the CP-specific antibodies were used in RT-PCR and Western blot, respectively.

**Figure 4 ijms-24-05477-f004:**
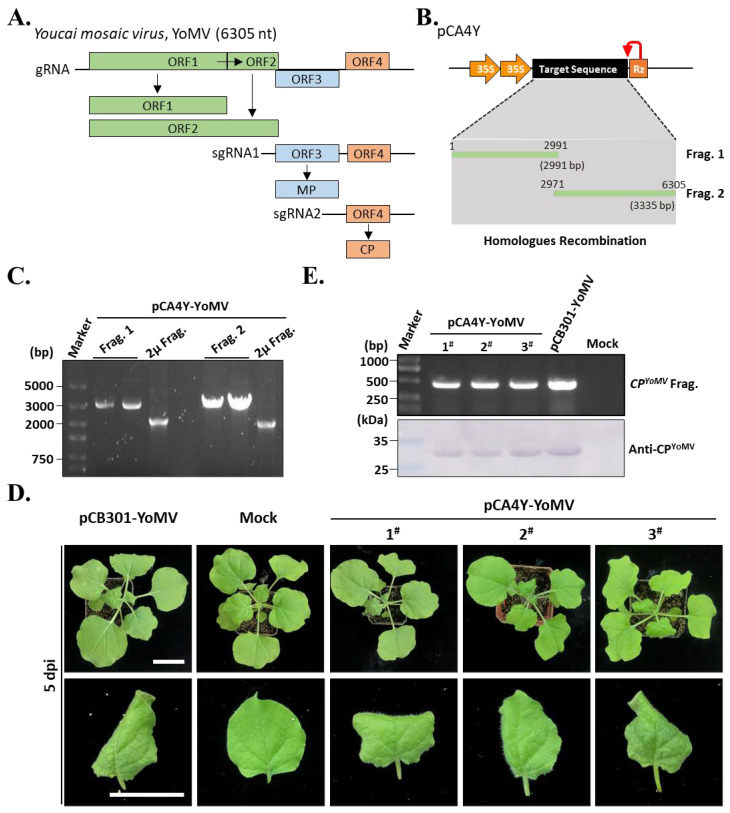
Construction of the *Youcai mosaic virus* (YoMV) infectious cDNA clone using the established triple-shutting vector-based yeast HR recombination. (**A**). Genome organization of the YoMV. Similarly, the ORF1 and ORF2 served as the viral replicase. The sub-genomic RNA1 and RNA2 encode the MP and CP. (**B**). Illustration of the detailed construction strategies for the YoMV single cDNA. The cDNA was divided into two parts: one is 2991 bp (Frag. 1), and another one is 3335 bp (Frag. 2). (**C**). PCR validation of the positive YoMV infectious cDNA clone with specific pairs of primers listed in Appendix A. The detailed meaning of the Frag. 1 and Frag. 2 were shown in (**B**). (**D**). Symptoms of the YoMV infectious cDNA clones-inoculated *N. benthamiana* plants after 5 dpi. The pCB301-YoMV-inoculated plants were treated as positive control. The mock, and numbers shown were the same meaning in Figure 3D. Bar indicates 3 cm. (**E**). RT-PCR and Western blot detect the infectivity of the newly constructed YoMV infectious clones. Specific pairs of primers and antibodies for the *CP* coding gene and protein were used in the assays, respectively.

**Figure 5 ijms-24-05477-f005:**
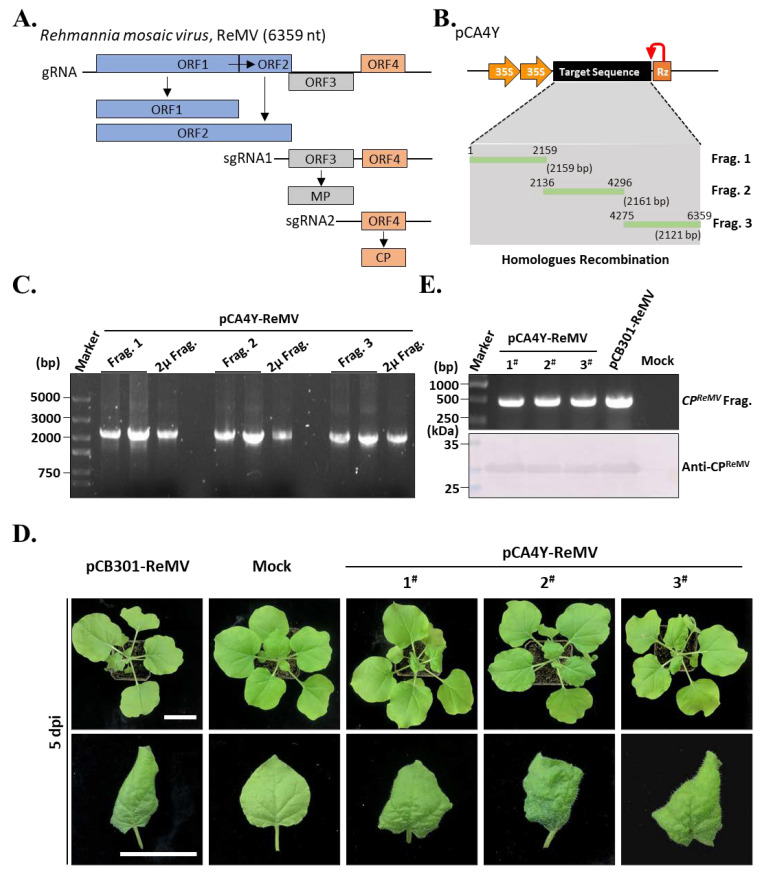
Construction of the *Rehmannia mosaic virus* (ReMV) infectious cDNA clone using the established triple-shutting vector-based yeast HR recombination. (**A**). Genome organization of the ReMV. The coding strategies were similar to the YoMV described in Figure 4A. (**B**). Illustration of the detailed construction strategies for the ReMV single cDNA. The cDNA was divided into three parts: fragments 1 is 2,159 bp (Frag. 1), fragment 2 is 2161 bp (Frag. 2), and fragment 3 is 2121 (Frag. 3). (**C**). PCR validation of the positive ReMV infectious cDNA clone with specific pairs of primers listed in Appendix A. The detailed meaning of the Frag. 1, Frag 2, and Frag. 3 were shown in (**B**). (**D**). Symptoms of the ReMV infectious cDNA clones-inoculated *N. benthamiana* plants after 5 dpi. The pCB301-ReMV-inoculated plants were treated as positive control. The mock, and numbers shown were the same meaning in Figure 4D. Bar indicates 3 cm. (**E**). RT-PCR and Western blot detect the infectivity of the newly constructed ReMV infectious clones. Specific pairs of primers and antibodies for the *CP^ReMV^* coding gene and protein were used in the assays, respectively.

## Data Availability

All data are available within the article and its Appendix A.

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
