# Peer review of "Generation of a Triple-Shuttling Vector and the Application in Plant Plus-Strand RNA Virus Infectious cDNA Clone Construction"

_ijms, 2023, doi:10.3390/ijms24065477_

Round 1

Reviewer 1 Report

The manuscript (Generation of A Triple-Shuttling Vector and the Application in Plant Plus-Strand RNA Virus Infectious cDNA Clone Construction) is interesting but I have some comments which should be addressed to improve the manuscript.

I recommend the author add short paragraphs in the introduction part about the economic impact of oilseed mosaic virus and Rehmannia mosaic virus

Please provide ELISA results.

I noted that the writing method of virus names are not uniform in the manuscript; please change the scientific names of the viruses in the manuscript by putting the first letter in capital and all the name in italic.

I recommend that you include a section in the materials and method on the isolation and purification of YoMV and ReMV.

L18 and 20 E. coli should be italic.

L20 -------- change to a high concentration

L21 -------- change to the construction

L23 -------- Please write E. coli and Agrobacterium tumefaciens in italic

L26 --------- change to infectious viral clones

L31 --------- This sentence was repeated in the Abstract (Plant virus infectious clones are a powerful tool for reverse genetic manipulation of viral genes in studying virus-host plant interactions and have contributed to a deeper understanding of virus life cycle and pathogenesis).

L95 --------- change to the construction

L 135-140 ----- Please re-write because this style should be in Materials and methods not results.

L196 -------- Grammatical error (were should be was).

L291 -------- change to E. coli are unstable

L293 -------- change to mutated easily if constructed

L360 -------- change to after the experimental test

I highly recommend you add clearer pictures of virus symptoms as the first lines of these figures already show leaf roll symptoms, but you can take pictures of the infected leaves between two layers of glasses to appear the mosaic symptoms.

Reviewer 2 Report

The manuscript by Feng et al describes a novel procedure to generate plant infectious cDNA clones. This research present novel information on how to make infectious cDNA clones of plant viruses which might be difficult to generate using standard gene cloning . The manuscript is well written, simple and clear.It represents a very useful tool for plant virology. I have only small comments to the authors:

Lines 102 – 105 : I believe the ending of the sentence is missing.

Lines 150-151 : I believe the authors should desbride more properly pCB301-CMV. It is just mentioned that they used it as a control. However, there is no description on what type of CMV infectious clone it is and why it needs to be grafted into the plant. 

Lines 338-339 : The authors must specify the infectious cDNA clone obtain from professor Xianbing

Round 2

Reviewer 1 Report

The authors have amended the manuscript according to our suggestions.
